# Effects of Surface-Type Plyometric Training on Physical Fitness in Schoolchildren of Both Sexes: A Randomized Controlled Intervention

**DOI:** 10.3390/biology11071035

**Published:** 2022-07-10

**Authors:** Hamza Marzouki, Rached Dridi, Ibrahim Ouergui, Okba Selmi, Rania Mbarki, Roudaina Klai, Ezdine Bouhlel, Katja Weiss, Beat Knechtle

**Affiliations:** 1High Institute of Sports and Physical Education of Kef, University of Jendouba, Kef 7100, Tunisia; hamzic_30@hotmail.com (H.M.); dridirached96@gmail.com (R.D.); ouergui.brahim@yahoo.fr (I.O.); okbaselmii@yahoo.fr (O.S.); rania.mbarki95@gmail.com (R.M.); roudainaklai100@gmail.com (R.K.); 2Laboratory of Cardio-Circulatory, Respiratory, Metabolic and Hormonal Adaptations to Muscular Exercise, Faculty of Medicine Ibn El Jazzar, University of Sousse, Sousse 4000, Tunisia; ezdine_sport@yahoo.fr; 3Institute of Primary Care, University of Zurich, 8006 Zurich, Switzerland; katja@weiss.co.com; 4Medbase St. Gallen Am Vadianplatz, 9100 St. Gallen, Switzerland

**Keywords:** physical activity, strength training, gender, strength fitness, endurance-intensive fitness, health status

## Abstract

**Simple Summary:**

Physical education classes seem to provide an excellent environment to promote health-related physical fitness development and an active lifestyle through the implementation of moderate-to-vigorous physical activity, such as plyometric training (PT). Researchers agree that youth PT approaches can provide a safe and effective conditioning strategy and should be an essential component of fitness, health promotion, and injury prevention programs. It could thus be important to investigate this shortfall within the context of physical education classes and athletic training prescriptions in the untrained school population of different sexes. Thus, the aim of this study was to compare the effects of short-term surface-type PT (firm vs. sand) on physical fitness performances in schoolchildren of both sexes. The data showed that both training surfaces presented greater pre-post changes in all assessed physical variables than non-plyometric programs, which is in concordance with with the accumulated evidence of PT’s effectiveness in improving youth health status. No significant differences in pre-post changes were observed relative to surface type and gender. Since many schools do not have access to sand surfaces, firm surfaces remain the recommendation when PT is envisaged for schoolchildren.

**Abstract:**

Plyometric training (PT) has been found to be effective for children’s fitness. However, no study has examined the effects of sex on physical fitness adaptations from surface-type PT in children. This study compared the effects of short-term surface-type PT (firm vs. sand) on the physical fitness of schoolchildren of both sexes. Sixty girls (age = 10.00 ± 1.15 years) and sixty boys (age = 10.02 ± 1.12 years) participated in a short-term (4 weeks), randomized and parallel PT design with pre-to-post measurements. Children were divided into two experimental groups (firm group: PT performed on a clay surface, 20 boys and 20 girls; sand group: PT performed on a dry surface of 20 cm deep sand, 20 boys and 20 girls) and a control group (CG, 20 boys and 20 girls). Squat jump, standing long jump, 20 m sprint, 5-10-5 shuttle, dynamic balance, and maximal aerobic velocity were measured at baseline and after intervention. Both experimental groups showed greater pre-post changes in all assessed variables than the CG (*p* < 0.0001). No significant differences in pre-post changes were observed relative to surface type or sex (*p* > 0.05). These findings suggest that a twice-weekly PT program induced physical fitness improvements, which may have transfer to health status during childhood. Additionally, surface type and sex did not affect the training-induced changes in physical fitness.

## 1. Introduction

Physical fitness is considered a determinant component for the maintenance, development and improvement of children’s health and quality of life [1]. Therefore, physical education classes provide an excellent environment to promote health-related physical fitness development and an active lifestyle through the implementation of moderate-to-vigorous physical activity, such as plyometric drills [1].

Plyometric ability involves multi-joint movements (i.e., leaping, hopping, skipping) that make use of the stretch-shortening cycle (SSC), in which muscles are stretched quickly (eccentric phase) before being immediately shortened (concentric phase) [2,3]. Despite previous concerns regarding the risk of injury that strength training could pose to children, researchers agree that youth plyometric training (PT) approaches can provide a safe and effective strategy for conditioning and should therefore be included in youth fitness, health promotion, and injury prevention programs [4,5,6,7]. Likewise, studies have shown health benefits from PT, notably improvements in vertical and horizontal jump performances, running speed, agility, change of direction speed (COD), balance ability, and endurance adaptations in children [1,2,8,9]. Hence, PT could be an effective modality to further improve children’s motor abilities, such as running, hopping, kicking, and throwing, all of which are commonly used in playing and recreational activities, as it generates dynamic motions and greater force in muscles and bones [10].

Plyometric training is commonly performed on hard surfaces [8,11,12], with the rationale that more compliant surfaces (e.g., sand) usually store the generated muscle energy and, hence, reduce the elastic rebound force [13]. In contrast, previous studies have suggested using a sand surface as an effective strategy to enhance neuromuscular performance, as it may potentially increase the activation of the stressed muscles in the target motor task [14,15,16]. In this context, findings from prior investigations have shown that the level of PT effect on physical fitness is influenced by the type of training surface (e.g., grass, sand, or firm) [17,18,19]. However, these studies were conducted only with young [17] or adolescent male athletes [18,19]. It is still unclear whether the surface type (firm vs. sand) can differently affect the plyometric training-induced performance adaptations (i.e., interventions ≥ 4 weeks) in untrained schoolchildren.

In addition to PT configuration, the subjects’ characteristics, such as sex, should be considered to optimize training-related responses [20]. Previous studies suggested that the age from 8 to 13 years can be considered as a sensitive period for the development of nervous and endocrine systems, as well as the corresponding anthropometric and physiological changes, which may affect the level of motor performance and their responses to learning and training stimuli [8,21,22,23]. Since sensitivity to PT programs may change differently according to sex over time/with maturation [4], we believe that it would be useful for practitioners to know whether sex can affect athletic performance following PT in prepubescent individuals. Specifically, only a few studies have examined sex-related effects following PT during the pre-peak high velocity (PHV) of children, and they did not detect any training-related differences between sexes in terms of sprinting, vertical and long jumps, or endurance [1,24,25]. Unfortunately, none of these studies compared the sexes’ agility and dynamic balance responses. To the best of our knowledge, no study has examined the effects of sex on physical fitness adaptations from surface-type PT in children.

Therefore, it could be important to investigate this shortfall within the context of physical education classes and athletic training prescriptions in the untrained school population of different sexes. Thus, the aim of this study was to compare the effects of short-term surface-type PT (firm vs. sand) on jump, linear running speed, agility, dynamic balance, and endurance-intensive performances in schoolchildren of both sexes. Given these premises, our first hypothesis was that both surface-types PT would positively influence physical fitness performances more than the habitual physical education training program (control group: CG). We also hypothesized that there would be no training-related differences between sex groups in the assessed variables after four weeks of PT.

## 2. Materials and Methods

### 2.1. Study Design

In this experimental study, we used a randomized and parallel-group design with pre- and post-measurement across four weeks of PT. The PT was performed after the warm-up in each physical education session. During the rest of the physical education session, the experimental groups joined the control groups and completed the same exercises. The control groups participated in their regular physical education lessons (i.e., gymnastics, relay games, games with a ball) during the PT intervention.

This study was conducted during the first trimester of the 2020–2021 school year (October–November), with children who completed two physical education sessions per week (1 h/session) and who had not been involved in any after-school activities or any other training programs (i.e., resistance, PT) for at least six months prior to the study. The study lasted six weeks and consisted of one week of pre-testing (T1), four weeks of PT (twice a week), and one week of post-testing (T2). Children were tested for muscular power (squat jump (SJ), standing long jump (SLJ)), linear running speed, agility, dynamic balance, and endurance-intensive performances.

One week before beginning the experiment, all children participated in two orientation sessions to become familiar with the experimental procedures and reduce the learning effect during the study. Each familiarization session was performed over two days, with 48 h of rest between the two sessions. Specifically, all children were familiarized with both testing procedures and the exercises included in the training program. Test results from the second familiarization session and T1 were also used to calculate the intraclass correlation coefficient (ICC). Children were asked not to perform any vigorous physical activity on the day before or on the day of any study procedures. All testing procedures were completed at the same time of day for both pre- and post-tests and under similar environmental conditions (indoors in the school gymnasium on the floor) to maintain a consistent surface and eliminate environmental stimuli commonly encountered outdoors. They were carried out over two days separated by ≈48 h of rest. Body height, leg length, and body mass were measured to the nearest 0.01 m and 0.1 kg using a digital scale (OHAUS, Florhman Park, NJ, USA). Body mass index (BMI) was calculated using the standard formula: BMI = weight (kg) × height^−2^ (m). Linear running speed, agility, and SJ tests were performed on the first day.

SLJ, dynamic balance, and endurance-intensive performances were assessed on the second day. Before the testing session, children completed a 15 min standardized warm-up session, consisting of 10 min of jogging and light stretching followed by five minutes of intense exercises (sets of short-sprints, skipping, and leg and arm swings). The muscular power, linear running speed, agility, and dynamic balance tests involved two valid maximal trials interspersed with two minutes of passive recovery, and the best performance was used for analysis. In the endurance-intensive test, only one maximal trial was performed. Five minutes of rest were allowed between tests. The same research assistants conducted all testing procedures and the training program. Verbal encouragement was provided to each child during all testing and training sessions.

### 2.2. Participants

The sample consisted of one hundred and thirty-seven school children aged 8 to 11.5 years (71 girls and 66 boys), all of whom volunteered for this investigation. Participants were from the same public primary school (Primary School, Kef city, Tunisia), had similar socio-economic status, and lived in the same city. Based on their weekly school timetable for physical education lessons, the children were divided into two experimental groups (firm surface group (FG): PT performed on a clay surface, *n* = 52; sand surface group (SG): PT performed on a dry surface of 20 cm deep sand, *n* = 45)) and a control group (CG, *n* = 40). To be included in the analysis, the children were required to complete at least 90% of the total training sessions. Due to low PT attendance, data for 17 children (11 girls and 6 boys) were excluded from further analysis. Consequently, 60 girls and 60 boys were included in the final analyses, including 40 from each group.

The exclusion criteria were: (1) having a disability or having experienced orthopedic injuries or surgeries that limited movement; (2) having comorbidities (i.e., cardiorespiratory failure, asthma); (3) having undertaken any other training programs (i.e., resistance, PT) during the period of the intervention; (4) having missed any of the testing sessions; (5) lack of parental consent for participation in the study. Anthropometric and demographic characteristics for the participants from different groups are included in Table 1. Children were thoroughly informed about the study design and, subsequently, their parents/legal guardians signed a written informed consent form prior to the start of the experimentation. Parents/legal guardians and children were informed that participation was voluntary and that they could withdraw from the study at any time. The protocol was conducted according to the Declaration of Helsinki concerning human research [26] and approved by a local research ethics committee (approval no. 013/2020).

### 2.3. Procedures

#### 2.3.1. Assessment of Muscular Power

We adopted the SJ test to assess the explosive lower-body power (height in cm). Children were tested for SJs, following Bosco et al. (1983) [27], by using an infrared jump system (Optojump; Microgate, Bolzano, Italy). Children began the test in a flexed position with a 90° knee flexion angle to perform a maximum upward motion. The hands were on the hips to avoid the involvement of the arms. Children had to jump as high as possible with extended knees and ankles and land where they took off.

Regarding SLJs, children were instructed to stand behind the start line with feet slightly apart, jump forward as far as possible and land on both feet without falling backwards. To achieve a vigorous jump, children were allowed to swing their arms and bend their knees. The distance from the start line to the first point of contact of the heels with the ground was measured in centimeters with a tape measure.

#### 2.3.2. Assessment of Linear Running Speed and Agility

To test linear running speed, time over a distance of 20 m was recorded using a series of paired photocells (Globus, Microgate, SARL, Italy). Children were instructed to start from a standing position, with the front foot placed 5 cm in front of the first photocell beam.

For agility, the 5-10-5 shuttle test was performed following Harman et al. (2000) [28]. From a neutral stance and straddling the start/finish line, children were instructed: (a) to turn and sprint to the right (4.55 m), then touch the cone with their right hand; (b) to turn to the left, run (9.10 m) to the far cone, and touch it with their left hand; (c) and to turn and sprint 4.55 m to the start/finish line. The time was recorded with a single photocell beam, with the photocells set perpendicular to the course at 0.30 m from the ground and positioned 1.3 m apart.

#### 2.3.3. Assessment of Dynamic Balance

To evaluate the dynamic balance, the lower quarter Y-balance test (YBT) was set up and administrated using the protocol outlined by Chaouachi et al. (2017) [29]. The composite score was calculated and used for subsequent analysis.

#### 2.3.4. Assessment of Endurance-Intensive Performance

We adopted the 20 m multistage shuttle run test [30] to determine the maximal aerobic velocity (MAV). Children were instructed to run back and forth between two lines 20 m apart with continuously increasing velocity, until they failed to reach the line in two consecutive shuttles or stopped voluntarily. Children started with an initial running velocity of 8 km·h^−1^, which increased by 0.5 km·h^−1^ each minute, with the required running speed in each sequence dictated by the beep sounds from an electronic audio recording. The maximal oxygen consumption (VO_2_max) was estimated by using the following equation [30]:VO_2_max (mL·kg^−1^·min^−1^) = 31.025 + 3.238X − 3.248A + 0.1536XA
where X = maximal aerobic velocity (km·h^−1^) and A = age (years).

### 2.4. Plyometric Training

The PT lasted four weeks, consisted of twice-weekly sessions on non-consecutive days, and was performed after the warm-up of the physical education sessions, which involved low-intensity aerobic activity and dynamic stretching of the lower limb muscles [31]. No pilot study was conducted to test the intervention program. The length of the training period was selected in line with previous research, where the effectiveness of a four-week PT program on SSC measures has been reported [31], and to allow for a test week before and after the training intervention within the educational term. Table 2 shows how the training program drills were based on findings from previous investigations [31] and incorporated the progressive overload principle by increasing the number of jumps and the levels of complexity according to previous PT guidelines [32]. The total number of contacts per session progressed from 64 to 104, and each session included 4–6 jump exercises (content detailed in Table 2). Each exercise lasted approximately 5–10 s, and at least 90 s of rest was allowed after each set. Both PT groups performed the same program during the intervention on different surfaces: the FG used a clay surface, while the SG used sand. Children were asked to exert maximum effort throughout the jumps, with minimal ground contact times. Whenever an exercise was not performed correctly, it was interrupted and repeated. During the training period, no damage or injuries occurred.

### 2.5. Statistical Analysis

Data are presented as means ± SD. The analyses were performed with SPSS version 20.0 for Windows (IBM Corp, Armonk, NY, USA), with a significance level of *p* ≤ 0.05. Relative reliability was assessed using the intraclass correlation coefficient. A sample *t*-test was performed to compare pre- and post-test data for each group. The pre-post change percentage (Δ%) was calculated for each physical variable and used for inferential testing. After testing the data normality (Shapiro–Wilk test) and homogeneity (Levene test), assumptions were checked, and a two-way univariate analysis of variance (sex × condition) was performed to assess sex effects on the variables’ percentage changes in response to surface-type plyometric training. When a significant F-value occurred for the interaction or main effect, Bonferroni post hoc procedures were performed. To determine the magnitude of differences, partial eta squared was converted to Cohen’s d effect size (ES) [33]. Threshold values for assessing magnitudes of ES were <0.20, 0.20–0.49, 0.50–0.79, and ≥0.80 for trivial, small, medium, and large, respectively [33]. G*Power (Version 3.1.9.4, University of Kiel, Kiel, Germany) was utilized to calculate the sample size power using the F test family (ANOVA: fixed effects, special, main effects, and interactions) with sex (boys and girls) × condition (firm (FG), sand (SG), or control (CG)). The analysis revealed that a total sample size of 100 children would be sufficient to find significant differences (effect size f = 0.40, *p* = 0.05, statistical power (1-β) = 0.95) with an actual power of 95.06%.

## 3. Results

The normality of the data and the homogeneity of the variance were confirmed. The ICCs for the test–retest trial were 0.961 for SJ, 0.887 for SLJ, 0.935 for the linear running speed, 0.913 for agility, 0.964 for Y-balance composite score, and 0.842 for MAV, indicating good to excellent agreement between trials. All physical performance variables improved after the training period in all groups (all *p* < 0.01) (Table 3 and Table 4). However, physical capabilities had similar improvements with both surfaces (all *p* > 0.05). The absolute values resulting from the analyses between and within groups are shown in Table 3 and Table 4.

### 3.1. Muscular Power

Trivial magnitude and non-statistically significant differences were observed between sexes for SJ (F = 0.039; *p* = 0.844, ES = 0) and SLJ (F = 1.880; *p* = 0.173, ES = 0.087). No statistical interactions were observed between sex and condition (SJ = F = 0.119; *p* = 0.888, ES = 0, and SLJ = F = 0.054; *p* = 0.948, ES = 0). However, a main effect for condition was identified, in which SJ (F = 133.397; *p* < 0.0001, ES = 1.504) and SLJ (F = 148.449; *p* < 0.0001, ES = 1.588) pre-post changes were higher in the SG and FG than the CG (SJ: 95% CI = 7.110 to 10.002 and 6.830 to 9.722, ES = 3.307 and 3.103, all *p* < 0.0001, respectively; SLJ: 95% CI = 7.763 to 10.766 and 7.674 to 10.677, ES = 3.510 and 3.509, all *p* < 0.0001, respectively) for both boys and girls (Table 3).

### 3.2. Linear Running Speed

Observing the linear running speed, no statistical interaction (F = 0.778; *p* = 0.462, ES = 0) or main effect for sex (F = 0.443; *p* = 0.507, ES = 0) were found. In contrast, a large magnitude and statistically significant main effect for condition were observed (F = 453.918; *p* < 0.0001, ES = 2.782), in which the SG and FG showed higher pre-post changes than the CG (95% CI = 2.725 to 3.279 and 2.661 to 3.214; ES = 5.692 and 6.559; all *p* < 0.0001, respectively) for both boys and girls (Table 3).

### 3.3. Dynamic Balance

For the dynamic balance performance, no statistical interaction (F = 0.167; *p* = 0.847, ES = 0) or main effect for sex (F = 0.357; *p* = 0.552, ES = 0) were found. In contrast, a large magnitude and statistically significant main effect for condition were observed (F = 71.497; *p* < 0.0001, ES = 1.098), in which the SG and FG showed higher pre-post changes than the CG (95% CI = 4.402 to 7.020 and 4.127 to 6.745; ES = 2.544 and 2.424; all *p* < 0.0001, respectively) for both boys and girls (Table 4).

### 3.4. Agility and VO_2_max

Regarding agility and VO_2_max performances, trivial magnitude and non-statistically significant differences were observed between sexes (F = 1.003 and 0.047; *p* = 0.319 and 0.829, ES = 0.005 and 0, respectively). No statistical interactions were observed between sex and condition (agility = F = 0.514; *p* = 0.599, ES = 0, and VO_2_max = F = 0.051; *p* = 0.950, ES = 0). However, a main effect for condition was identified, in which agility (F = 461.160; *p* < 0.0001, ES = 2.805) and VO_2_max (F = 76.340; *p* < 0.0001, ES = 1.135) pre-post changes were higher in the SG and FG than the CG (agility: 95% CI = 4.683 to 5.634 and 4.658 to 5.609; ES = 6.248 and 5.852; all *p* < 0.0001, respectively; VO_2_max: 95% CI = 5.848 to 9.145 and 5.350 to 8.647; ES = 2.473 and 2.339; all *p* < 0.0001, respectively) for both boys and girls (Table 4).

## 4. Discussion

This study aimed to compare the effects of short-term surface-type PT (firm vs. sand) on physical fitness performances in schoolchildren of both sexes. In agreement with our hypothesis, our findings showed that both the FG and SG demonstrated more significant improvements in SJ, SLJ, linear running speed, agility, dynamic balance, and endurance-intensive performances than the CG. Furthermore, the assessed physical abilities had similar improvements with both surfaces. Moreover, our results revealed that sex had no significant impact on pre-post variations in any of the measured physical variables, which confirms our second hypothesis.

Previous studies have highlighted PT’s importance in improving jumping ability [1,7,8,9,34,35]. In fact, it has been reported that interventions using intensive hopping exercises are generally favorable for the optimization of children’s jump ability [10]. Likewise, the results of the present study support the idea that including a PT modality is a recommendable approach to enhance jump performance in a short period. Thus, the improvements in plyometric-induced jumping performance found in all experimental groups could be related to the stretch reflex mechanism adaptations and the muscle’s ability to store elastic energy [36]. Those adaptations manifest as stronger and faster elastic recoil generated by greater muscle stiffness during ground contact [37]; earlier activation of the stretch reflex, resulting in greater activation of muscle fibers [38]; better use of elastic energy [38]; and a decrease in Golgi tendon organ activity [38,39,40].

Our findings demonstrated that all PT groups achieved similar improvements in linear running speed and agility performances, which were greater than their non-plyometric training counterparts. Prior reports have noted performance gains in linear speed and agility following PT [1,7,8,9,34]. Thus, the additional gains in linear speed and agility recorded in our experimental groups compared to the CG could be explained by an improvement in ground contact time (especially during acceleration) and musculotendinous stiffness [41,42,43,44]. Furthermore, the plyometric exercise protocol included several explosive lateral motions that may have improved the eccentric strength of the extensor muscles of the legs, which is a requirement to change direction while decelerating [45]. Neuromuscular adaptations that allow athletes to transition between deceleration and acceleration actions have also been related to PT’s development of agility [46].

In addition, our findings extend previous research that observed significant improvements in balance performance in response to PT [47,48]. Compared to the control training, PT resulted in additional gains in balance performance in all experimental groups. These results suggest that training programs using plyometric exercises can be a useful tool to enhance balance performance. These additional gains could be explained in part by improvements in neuromuscular control [49], anticipatory adjustments [50], and the sensitivity of the sensory feedback tract during exercise [51]. In the lower limbs, PT induces several neuromuscular adaptations [52], such as improvements in muscle structure and functional behavior (e.g., muscle size and architecture, changes in musculotendinous stiffness and inter-muscular coordination) [36], which could enhance motor performance and reduce the risk of injury [53].

Our data also showed a significant improvement in endurance-intensive performance, highlighting the potential gains from applying plyometrics in U12 untrained children. These results extend previous research that observed significant improvements in VO_2_max performance in response to PT in prepubescent children [1,48]. The high increases in VO_2_max in all experimental groups compared with the CG suggest that the implementation of a school-based PT program can be a positive stimulus to improve endurance in healthy schoolchildren. Thus, the plyometric training-induced VO_2_max increases might have related to a better running economy [54], increased musculotendinous stiffness, and neuromechanical improvements [36]. These can allow a faster transfer of force from contracting muscles to moving bones via tendons [55], reducing reaction times [56]. However, further studies considering the direct assessment of the potential mechanisms involved in improving endurance-intensive performance after PT are warranted.

Children aged 8–13 years pass through a dynamic developmental period that is marked by rapid changes in anthropometric and physiological parameters, and it can affect their level of motor performance, as well as their responses to learning and training stimuli [8,21,22,23]. In the present study, both the experimental and control groups presented significant pre-to-post improvements in all physical tests. However, the magnitude of the improvement was greater in the PT groups. Considering the effect of PT, our findings suggest that conditioning exercises focusing intensively on improvements in physical skills are more propitious in evincing children’s success [10]. Thus, the additional gains found in PT groups might be related to biomechanical parameters (e.g., contractile and elastic musculoskeletal properties, maximal isometric voluntary force, musculotendinous stiffness) [57]. Furthermore, training programs including motions that are biomechanically and metabolically specific to the performance testing are beneficial for further improving athletic performance [5,58], and thus might represent an appropriate stimulus to stress the physical abilities of interest. Moreover, it can be inferred that the additional gains observed in PT groups were induced by the utilized training load.

The present study showed similar physical performance improvements after PT with sand and firm surfaces. These findings are in line with previous reports indicating that short- to middle-term PT performed on an unstable surface (i.e., sand) produced similar linear running speed, jump, agility, balance, and endurance performance improvements as those observed after training on stable surfaces (i.e., firm) [3,12,14,17,18,19]. In this regard, it has been reported that performing explosive tasks (i.e., sprinting, jumping) on firm surfaces may improve the muscles’ efficiency in utilizing the elastic energy stored during the eccentric phases, generating powerful concentric actions [12,16,59,60]. Thus, the plyometric training-induced adaptations on firm surfaces may be attributed to increased efficiency in storing and reusing elastic energy in explosive motions [14]. In contrast, previous studies have reported that sand training may increase the level of muscle activation and energy cost resulting from a considerable amount of elastic energy dissipation [61,62], and it could serve as an alternative way to increase overload during workouts [14]. As both strategies are easily implemented and efficient in improving physical fitness performances, it is possible that low- and moderate-volume sessions of these two distinct and possibly complementary mechanisms could be planned before sport-specific activities, even as warm-up strategies [14].

On the other hand, the present study did not detect any training-related differences between sexes for any of the assessed variables after short-term PT performed on either firm or sand surfaces. These findings are in line with previous research indicating that PT responses are not affected by sex in prepubescent children [1,4,24,25,63]. Regardless of maturity levels, children in this study attended the same school and participated in the same physical education programs. It has been reported that the physical fitness responses to training could be potentially related to biological maturation in body size, shape, and composition [1]. Thus, this knowledge should be considered to optimize well-rounded school-training programs at times of rapid changes, such as the prepubertal growth spurt [1]. Despite the maturation process occurring earlier in boys, girls gain around 3.5 kg of muscle mass annually and accumulate a higher amount of fat mass during PHV, while boys gain much more muscle (7.2 kg per year) [64]. These concurrent processes can disproportionately reduce strength in girls [65]. Unfortunately, the current study lacked the further physiological and biomechanical assessments that would help better understand the underlying mechanisms of plyometric training-induced adaptations in both sexes. Information relating to these factors may help practitioners to optimize well-rounded school-training programs at times of rapid changes, such as the prepubertal growth spurt [1].

Finally, some limitations should be listed for the present investigation: (i) the training period was short, whereas a longer period may be required to observe greater differences in physical fitness performances between the different plyometric training conditions and/or sexes. (ii) The study lacked further physiological and biomechanical assessments that would help better understand the underlying mechanisms of plyometric training-induced adaptations in both boys and girls during childhood. Thus, developmental characteristics should be considered when planning children’s physical exercise protocols in order to respect neuromotor plasticity [8]. (iii) Since many schools do not have access to sand surfaces and the present study did not find statistical differences between the two surfaces, it would be difficult to suggest the use of sand for a population of schoolchildren, and firm surface remains the recommendation when PT is envisaged for schoolchildren. (iv) Finally, no questionnaire was adopted to identify sedentary levels in our study. However, to our knowledge, there is no validated self-reported instrument in the Arabic language that assesses sedentary behavior among children and relates sedentary time to social, environmental, and health outcomes.

## 5. Conclusions

Implementing a four-week PT program during physical education sessions as a substitute for some physical education exercises induced higher jump, linear running speed, agility, balance, and endurance performance enhancements than non-plyometric training among untrained schoolchildren. Thus, the incorporated biweekly PT program induced physical fitness enhancements, which may transfer to health status during childhood. Although both training surfaces (sand and firm) showed similar improvements in all assessed variables, the sand training program can be considered an effective training modality that can be incorporated into weekly physical education routines in conjunction with plyometrics on harder surfaces. However, the improvements generated by PT were not influenced by gender. Further research elucidating gender adaptations following PT at different age stages is warranted.

## Figures and Tables

**Table 1 biology-11-01035-t001:** Anthropometric and demographic characteristics for the experimental and control groups. (*n* = 120).

Groups	Sex (*n*)	Age (Years)	Height (cm)	LL (cm)	Weight (kg)	BMI (kg·m^−2^)
FG	Boys (20)	10.1 ± 1.2	143.1 ± 10.6	70.9 ± 5.2	36.1 ± 9.8	17.3 ± 2.8
	Girls (20)	10.0 ± 1.1	143.3 ± 12.7	71.0 ± 6.3	38.1 ± 10.6	18.3 ± 3.3
SG	Boys (20)	10.0 ± 1.2	142.2 ± 9.86	70.2 ± 4.7	35.4 ± 7.8	17.3 ± 2.0
	Girls (20)	10.0 ± 1.1	143.0 ± 10.4	70.7 ± 5.2	37.4 ± 8.5	18.0 ± 2.1
CG	Boys (20)	10.0 ± 1.2	141.7 ± 11.0	70.0 ± 4.9	35.7 ± 8.4	17.5 ± 2.0
	Girls (20)	10.1 ± 1.1	143.5 ± 9.83	70.9 ± 5.6	38.8 ± 11.7	18.5 ± 3.4

Values are given as means ± SD; FG: firm group; SG: sand group; CG: control group; LL: leg length; BMI: body mass index.

**Table 2 biology-11-01035-t002:** Description of the plyometric program.

	Week 1		Week 2		Week 3		Week 4	
Type of Jumps	Session 1	Session 2	Session 1	Session 2	Session 1	Session 2	Session 1	Session 2
Pogo jump	2 × 6	2 × 6	2 × 8	2 × 10	2 × 10	4 × 8	4 × 8	4 × 10
Lateral jump	2 × 6	4 × 6	2 × 8					
Hop scotch	3 × 4							
Bilateral power hops	4 × 4	4 × 4	4 × 4					
Ankle hops	2 × 6	3 × 5	3 × 5	3 × 5				
Power skipping			2 × 6	2 × 8	3 × 8			
Unilateral pogo jump				2 × 8	2 × 10	2 × 8	2 × 8	2 × 10
Max rebound hops				3 × 5	3 × 5	3 × 5	4 × 5	
Drop jump					2 × 5	2 × 5	2 × 5	2 × 6
Hurdle power hops						2 × 6	2 × 5	2 × 5
Double tuck jumps						2 × 5	2 × 6	2 × 6
Alternating jump lunges								2 × 5
Total foot contacts	64	67	75	82	89	95	100	104

Number of sets × number of repetitions; 90 s of passive recovery between sets.

**Table 3 biology-11-01035-t003:** Linear running speed (s) and jump (cm) performances at baseline and after the intervention period for all groups (*n* = 120).

Variables	Group	Sex (*n*)	Pre-Test	Post-Test	Δ (%)	*p*-Value (ES)	95% CI
S20	Boys	FG (20)	4.31 ± 0.41	4.14 ± 0.40 ^†^	−4.0 ± 0.4	FG vs. CG: <0.0001 (7.319) SG vs. CG: <0.0001 (6.789)	FG vs. CG: 2.552–3.335 SG vs. CG: 2.737–3.519
		SG (20)	4.32 ± 0.28	4.14 ± 0.27 ^†^	−4.2 ± 0.5		
		CG (20)	4.33 ± 0.25	4.28 ± 0.25 ^†^	−1.0 ± 0.4		
	Girls	FG (20)	4.57 ± 0.33	4.38 ± 0.33 ^†^	−4.2 ± 0.5	FG vs. CG: <0.0001 (5.996) SG vs. CG: <0.0001 (4.861)	FG vs. CG: 2.540–3.322 SG vs. CG: 2.485–3.267
		SG (20)	4.60 ± 0.32	4.41 ± 0.32 ^†^	−4.1 ± 0.7		
		CG (20)	4.64 ± 0.30	4.59 ± 0.30 ^†^	−1.2 ± 0.5		
SJ	Boys	FG (20)	14.6 ± 2.7	16.3 ± 2.9 ^†^	11.7 ± 2.4	FG vs. CG: <0.0001 (3.652) SG vs. CG: <0.0001 (3.563)	FG vs. CG: 6.515–10.605 SG vs. CG: 6.604–10.694
		SG (20)	14.8 ± 2.5	16.5 ± 2.6 ^†^	11.8 ± 2.6		
		CG (20)	14.4 ± 2.1	14.9 ± 2.4 ^†^	3.1 ± 2.3		
	Girls	FG (20)	13.4 ± 2.6	14.8 ± 2.8 ^†^	11.5 ± 3.1	FG vs. CG: <0.0001 (2.655) SG vs. CG: <0.0001 (3.028)	FG vs. CG: 5.946–10.037 SG vs. CG: 6.418–10.509
		SG (20)	13.0 ± 2.3	14.6 ± 2.3 ^†^	11.9 ± 2.6		
		CG (20)	13.5 ± 2.3	13.9 ± 2.1 ^†^	3.5 ± 3.0		
SLJ	Boys	FG (20)	81.9 ± 13.2	93.2 ± 14.4 ^†^	14.0 ± 3.0	FG vs. CG: <0.0001 (3.631) SG vs. CG: <0.0001 (3.724)	FG vs. CG: 7.247–11.493 SG vs. CG: 7.287–11.534
		SG (20)	82.8 ± 14.3	94.3 ± 15.7 ^†^	14.1 ± 3.0		
		CG (20)	81.8 ± 13.0	85.5 ± 13.0 ^†^	4.6 ± 2.0		
	Girls	FG (20)	72.6 ± 13.4	82.0 ± 14.5 ^†^	13.2 ± 2.9	FG vs. CG: <0.0001 (3.364) SG vs. CG: <0.0001 (3.285)	FG vs. CG: 6.858–11.105 SG vs. CG: 6.995–11.242
		SG (20)	73.2 ± 11.3	82.7 ± 11.7 ^†^	13.3 ± 3.1		
		CG (20)	73.3 ± 8.9	76.4 ± 9.9 ^†^	4.2± 2.4		

Values are given as means ± SD; S20: 20 m linear running speed; SJ: squat jump; SLJ: standing long jump; FG: firm group; SG: sand group; CG: control group; Δ (%): pre-post change percentage; ES: effect size; 95% CI: 95% confidence interval. ^†^ A significant difference when comparing pre-test to post-test. The statistical significance level was set at *p* ≤ 0.05.

**Table 4 biology-11-01035-t004:** Agility (s) and maximal oxygen consumption (mL·kg^−1^·min^−1^) performances, along with the composite score for the Y-balance test (%) at baseline and after the intervention period for all groups (*n* = 120).

Variables	Group	Sex (*n*)	Pre-Test	Post-Test	Δ (%)	*p*-Value (ES)	95% CI
Agility	Boys	FG (20)	7.40 ± 0.46	6.89 ± 0.49 ^†^	−7.0 ± 1.0	FG vs. CG: 0.0001 (5.982) SG vs. CG: 0.0001 (5.982)	FG vs. CG: 4.660–6.005 SG vs. CG: 4.580–5.925
		SG (20)	7.29 ± 0.52	6.78 ± 0.53 ^†^	−6.9 ± 1.0		
		CG (20)	7.29 ± 0.64	7.17 ± 0.63 ^†^	−1.7 ± 0.8		
	Girls	FG (20)	7.78 ± 0.56	7.26 ± 0.48 ^†^	−6.7 ± 0.9	FG vs. CG: 0.0001 (5.698) SG vs. CG: 0.0001 (6.423)	FG vs. CG: 4.263–5.608 SG vs. CG: 4.392–5.736
		SG (20)	7.72 ± 0.54	7.20 ± 0.52 ^†^	−6.8 ± 0.8		
		CG (20)	7.73 ± 0.46	7.60 ± 0.45 ^†^	−1.7 ± 0.8		
VO_2_max	Boys	FG (20)	42.4 ± 2.8	46.0 ± 2.8 ^†^	8.5 ± 3.0	FG vs. CG: 0.0001 (2.478) SG vs. CG: 0.0001 (2.632)	FG vs. CG: 4.714–9.377 SG vs. CG: 5.372–10.035
		SG (20)	42.1 ± 2.8	46.0 ± 2.6 ^†^	9.2 ± 3.2		
		CG (20)	42.0 ± 3.0	42.6 ± 3.0 ^†^	1.5 ± 2.6		
	Girls	FG (20)	42.4 ± 3.1	46.1 ± 3.4 ^†^	8.7 ± 2.9	FG vs. CG: 0.0001 (2.170) SG vs. CG: 0.0001 (2.275)	FG vs. CG: 4.620–9.283 SG vs. CG: 4.958–9.621
		SG (20)	42.3 ± 2.7	46.1 ± 3.0 ^†^	9.1 ± 2.9		
		CG (20)	42.0 ± 3.3	42.7 ± 3.2 ^†^	1.8 ± 3.5		
YBT	Boys	FG (20)	87.8 ± 6.7	94.7 ± 6.4 ^†^	8.1 ± 2.3	FG vs. CG: 0.0001 (2.624) SG vs. CG: 0.0001 (2.762)	FG vs. CG: 3.474–7.176 SG vs. CG: 3.552–7.255
		SG (20)	87.5 ± 5.2	94.5 ± 4.8 ^†^	8.1 ± 2.2		
		CG (20)	86.9 ± 6.1	89.2 ± 6.3 ^†^	2.7 ± 1.7		
	Girls	FG (20)	87.9 ± 4.6	94.8 ± 6.0 ^†^	7.7 ± 3.0	FG vs. CG: 0.0001 (2.245) SG vs. CG: 0.0001 (2.371)	FG vs. CG: 3.696–7.398 SG vs. CG: 4.167–7.869
		SG (20)	87.1 ± 4.4	94.3 ± 5.7 ^†^	8.2 ± 3.1		
		CG (20)	86.8 ± 3.1	88.7 ± 4.1 ^†^	2.2 ± 1.8		

Values are given as means ± SD; VO_2_max: maximal oxygen consumption; YBT: Y-balance test; FG: firm group; SG: sand group; CG: control group; Δ (%): pre-post change percentage; ES: effect size; 95% CI: 95% confidence interval. ^†^ A significant difference when comparing pre-test to post-test. The statistical significance level was set at *p* ≤ 0.05.

## Data Availability

The data presented in this study are available on request from the corresponding author. The data are not publicly available due to privacy reasons.

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
