# Peer review of "Effects of Surface-Type Plyometric Training on Physical Fitness in Schoolchildren of Both Sexes: A Randomized Controlled Intervention"

_biology, 2022, doi:10.3390/biology11071035_

Round 1
Reviewer 1 Report
The article deals with the important topic of children's physical fitness, which is one of the most important elements to ensure health. This topic assumes particular importance at a time when, as a result of the pandemic, children's activity and physical fitness have been severely limited.
The authors introduce the reader to the described problem in a clear and precise way. The part concerning the research methodology and the presentation of results are not objectionable.
I have only a few minor comments:
Line 103. You wrote that each subject attended two familiarization sessions the week before the experiment. Did each familiarization session also cover 2 days, as did the testing sessions? Were subjects familiarized with both the testing procedure and the exercises included in the training program during these sessions?
Line 105. What does the symbol "T1" mean? It was not explained earlier.
Line 175. It should be “in front of…” rather than “behind…”.
Line 197. Replace "year" with "years".
Table 2. Replace "Bilaterl hops power" with "Bilateral power hops". What did you mean by "Jumping lunge alternative"?
Author Response
Referee 1
Main comments:
The article deals with the important topic of children's physical fitness, which is one of the most important elements to ensure health. This topic assumes particular importance at a time when, as a result of the pandemic, children's activity and physical fitness have been severely limited.
The authors introduce the reader to the described problem in a clear and precise way. The part concerning the research methodology and the presentation of results are not objectionable.
Author's response:
We thank the expert reviewer for his/her comments. The comments were addressed in specific points.
Comment 1
Line 103. You wrote that each subject attended two familiarization sessions the week before the experiment. Did each familiarization session also cover 2 days, as did the testing sessions? Were subjects familiarized with both the testing procedure and the exercises included in the training program during these sessions?
Author's response:
Thank you for your valuable comment. The following changes are included in the study design section as follows:
'' Each familiarization session was performed over two days, with 48 h of rest between the two sessions. Specifically, all the subjects were familiarized with both the testing procedure and the exercises included in the training program.''
Comment 2
Line 105. What does the symbol "T1" mean? It was not explained earlier.
Author's response:
Thank you for your valuable comment.
T1 means Pre-test. Please find changes in the text.
Comment 3
Line 175. It should be “in front of…” rather than “behind…”.
Author's response:
We thank the expert referee for his/her comment.
''behind'' was replaced by ''in front of''. Please find changes in the text.
Comment 4
Line 197. Replace "year" with "years".
Author's response:
We thank the expert referee for his/her comment.
''year'' was changed by ''years''. Please find changes in the text.
Comment 5
Table 2. Replace "Bilaterl hops power" with "Bilateral power hops". What did you mean by "Jumping lunge alternative"?
Author's response:
We thank the expert referee for his/her comment.
"Bilaterl hops power" was replaced by "Bilateral power hops"
"Jumping lunge alternative" means ''Alternating jump lunges''. All changes were addressed and highlighted in the text.
Reviewer 2 Report
The study aimed to understand if surface typology may influence physical fitness in children undergoing plyometric training. Despite the study is well written and adds content regarding the effects of training in a population of school children I may question, is the journal and the special issue the best fit for this manuscript? It does not seem that the manuscript relates to biology or the biological underlying processes of physical exercise benefits.
However, below a point-by-point feedback for the authors:
Line 37, it would be advisable to include or modify the physical fitness components to the adopted evaluation tests since in the next line the authors refer to the “assessed variables”.
Lines 81-83 and lines 70-71 both describe the aim of the study. The authors should remove one of these sentences.
The introduction could add further background to the rationale of the study.
Lines 98 how did the authors control the involvement of the children in other physical education settings?
I assume that the authors provided the intervention during the physical education sessions. Therefore, how did they manage the children who were excluded due to exclusion criteria?
Was a questionnaire adopted to identify sedentary levels?
The agility test was not quite clear during the explanation, I would suggest the authors to re-write the paragraph.
Table 3 and 4 present data regarding the effects of PT. It is also important to note that the CG increased physical fitness. At this age it is very difficult to determine if components of physical fitness increase due to growth and development or to the activity proposed. However, magnitude of improvement is greater in the PT groups. Such aspect has been discussed in one sentence in the discussion, i would suggest to further stress this aspect.
The authors suggest that sand training may be preferable. Despite magnitude results greater for sand training there are two points which may be added as limitations. First, no statistical difference was evinced between sand and ground, and the second point is that practically, not many schools have sand surfaces, therefore it would result very difficult to suggest the use of sand in a population of school children.
In addition, the authors conclude that sex should be considered as a determinant variable, however this aspect is not evinced in the proposed results.
Thanks
Author Response
Referee 2
We thank the expert reviewer for his/her comments.
Comment 1
Line 37, it would be advisable to include or modify the physical fitness components to the adopted evaluation tests since in the next line the authors refer to the “assessed variables”.
Author's response:
We thank very much the expert referee for his/her comment.
All recommendations were added. Please find changes in the text.
Comment 2
Lines 81-83 and lines 70-71 both describe the aim of the study. The authors should remove one of these sentences.
Author's response:
We thank the expert referee for his/her comment.
Lines 70-71 were removed. Please find changes in the text.
Comment 3
The introduction could add further background to the rationale of the study.
Author's response:
We thank very much the expert referee for his/her comment.
Further background was added to the rationale of the study. Please find changes in the text.
Comment 4
Lines 98 how did the authors control the involvement of the children in other physical education settings?
Author's response:
We thank very much the expert referee for his/her comment.
Due to the pandemic (Covid-19) effects mainly on life behavior, children activity and participation in physical fitness ‘classes have been severely limited. In fact, according to the parents/caregivers declarations, children did not practice any physical activity (including strength and conditioning training programs) either inside or outside the schools (e.g., sports clubs and organized sports activities) from 15-03 until 15-09-2020.
Comment 5
I assume that the authors provided the intervention during the physical education sessions. Therefore, how did they manage the children who were excluded due to exclusion criteria?
Author's response:
We thank very much the expert referee for his/her comment.
After talking with the school administrators and doctor, we were asked to exclude children with health issues (who already did not perform any school activities) to avoid any health problem. However, the children who had missed any testing sessions or had not given parental consent for participation in the study were involved in their regular physical education sessions with the control group and were excluded from further analysis, as mentioned in the text.
Comment 6
Was a questionnaire adopted to identify sedentary levels?
Author's response:
We thank very much the expert referee for his/her question.
Unfortunately, no questionnaire was adopted to identify sedentary levels in our study. Sedentary behaviors are very prevalent nowadays. Objective measurement of sedentary behaviors is costly, requires technical expertise, and is challenging in terms of time and management. However, to our knowledge, there is no validated self-reported instrument in the Arabic language that assesses sedentary behavior among children and relates sedentary time to social, environmental, and health outcomes.
This can be added as a limitation of the study.
Comment 7
The agility test was not quite clear during the explanation, I would suggest the authors to re-write the paragraph.
Author's response:
We thank very much the expert referee for his comment. The agility test description was rewritten and added to the text as follows:
'' For agility, the 5-10-5 shuttle test was performed according to Harman et al. (2000). From a neutral stance and straddling the start/finish line, children were instructed: (a) to turn and sprint to the right (4.55 m), touching the cone with their right hand; (b) to turn to the left and run 9.10 m to the far cone, touching it with their left hand; (c) and to turn and sprint 4.55 m to the start/finish line. The time was recorded by a single photocell beam, set perpendicular to the course, at 0.30 m from the ground and positioned 1.3 m apart.''
Comment 8
Table 3 and 4 present data regarding the effects of PT. It is also important to note that the CG increased physical fitness. At this age it is very difficult to determine if components of physical fitness increase due to growth and development or to the activity proposed. However, magnitude of improvement is greater in the PT groups. Such aspect has been discussed in one sentence in the discussion, i would suggest to further stress this aspect.
Author's response:
We thank very much the expert referee for his/her comment. Further explanations about the magnitude of changes due to training interventions were added in the discussion section as follows:
'' Children aged 8-13 years are going through a dynamic developmental period that is marked by rapid changes in anthropometric and physiological parameters, and it could affect their level of motor performance as well as their responses to learning and training stimuli (Almeida et al., 2021; Degache et al., 2010; Duzgun et al., 2011; Sunnegardh et al., 1988). In the present study, both experimental and control groups presented significant pre-to-post improvements in all physical tests. In contrast, the magnitude of improvement was greater in the PT groups. Therefore, a more important way to interpret our results is related to the main effect of PT. Thus, when we consider the effect of PT, our data suggest that training programs focusing intensively on improvements in physical skills are more propitious to demonstrate success for children (Johnson et al., 2011). In this context, the improvements in plyometric-induced performance could be more related to biomechanical parameters (e.g., contractile and elastic musculoskeletal properties, maximal isometric voluntary force, musculotendinous stiffness) (Grosset et al., 2009). Furthermore, training programs including movements that are biomechanically and metabolically specific to the performance tasks may be likely to induce improvements in performance measures (Behm et al., 2017; Nobre et al., 2017), could represent therefore a suitable conditioning stimulus to stress the athletic components of interest. Along the same line, it is plausible to suggest that the training load utilized in PT groups was sufficient to induce additional gains to those observed in CG. ''
Almeida, M.B.; Leandro, C.G.; Queiroz, D.D.R.; José-da-Silva, M.; Pessôa Dos Prazeres, T.M.; Pereira, G.M. das-Neves, G.S.; Carneiro, R.C.; Figueredo-Alves, A.D.; Nakamura, F.Y.; Henrique, R.D.S.; Moura-Dos-Santos, M.A. Plyometric training increases gross motor coordination and associated components of physical fitness in children. Eur. J. Sport Sci. 2021, 21, 1263-1272. doi: 10.1080/17461391.2020.1838620.
Degache F, Richard R, Edouard P, Oullion R, Calmels P. 2010. The relationship between muscle strength and physiological age: a crosssectional study in boys aged from 11 to 15. Ann Phys Rehabil Med 53: 180–188.
Duzgun I, Kanbur NO, Baltaci G, Aydin T. 2011. The effect of tanner stage on proprioception accuracy. J Foot Ankle Surg 50:11–15.
Sunnegardh J, Bratteby LE, Nordesjo LO, Nordgren B. 1988. Isometric and isokinetic muscle strength, anthropometry and physical activity in 8 and 13 years old Swedish children. Eur J Appl Physiol 58:291–297.
Grosset, J.-F., Piscione, J., Lambertz, D., & Pérot, C. (2009). Paired changes in electromechanical delay and musculo-tendinous stiffness after endurance or plyometric training. European Journal of Applied Physiology, 105(1), 131–139.
Behm, D. G., Young, J. D., Whitten, J. H., Reid, J. C., Quigley, P. J., Low, J.,…Granacher, U. (2017). Effectiveness of traditional strength vs. power training on muscle strength, power and speed with youth: A systematic review and meta-analysis. Frontiers in Physiology, 8, 423.
Nobre, G. G., de Almeida, M. B., Nobre, I. G., dos Santos, F. K., Brinco, R. A., Arruda-Lima, T. R.,…Moura-dos-Santos, M. A. (2017). Twelve weeks of plyometric training improves motor performance of 7-to 9-year-old boys who were overweight/obese: A randomized controlled intervention. Journal of Strength and Conditioning Research, 31(8), 2091–2099.
Comment 9
The authors suggest that sand training may be preferable. Despite magnitude results greater for sand training there are two points which may be added as limitations. First, no statistical difference was evinced between sand and ground, and the second point is that practically, not many schools have sand surfaces, therefore it would result very difficult to suggest the use of sand in a population of schoolchildren.
Author's response:
We thank very much the expert referee for his/her comment. All points raised were added as limitations as follows:
''Since many schools do not have access to sand surfaces and that the present study did not find statistical differences between the two surfaces, it would be difficult to suggest the use of sand in a schoolchildren population, and that firm surface remains suggestible when PT is envisaged for schoolchildren.''
Comment 10
In addition, the authors conclude that sex should be considered as a determinant variable, however this aspect is not evinced in the proposed results.
Author's response:
We thank very much the expert referee for his/her comment.
As in the results’ section no main effect of sex or interaction effect between sex and conditions were found, it is not suitable to suggest that sex is a determinant variable that should be considered when planning PT in school children.
The suggestion was removed.
Reviewer 3 Report
Congratulations to the authors for the topic under study. However, this review study presents some issues that may need to be addressed. In the comments below, some considerations are set out.
Although the title informs the variables under study it is necessary to consider the existence of a high variety of research methods. Once the title is of vital importance for the selection and reading of research reports, it is recommended to identify the report as to the research design used, being coherent with the methodological procedures described.
The introductory section is well organized, including a contextualization of the problem under study, as well as a brief summary of the related state of the art. However, it is suggested that the rationale for the hypothesis ("We also hypothesized training-related differences between sex groups in the assessed variables after four weeks of PT.") could be structured in such a way as to inform the reader of its logic and consistency with a different depth, alluding to premises, concepts or theoretical models that are related to the object of study.
The "Participants" section is well described. However, since sample size plays an important role in the ability to make accurate inferences, was a statistical procedure performed to calculate the sample size? If so, it is suggested to report. Also, it seems unclear what sampling strategy was used. What was the procedure used for the participants to be allocated to the respective groups (experimental and control)? It is suggested to clarify.
The success of an intervention program also depends on the coach/professor who applies it. In the manuscript it is not clear if it was the same coach/professor who applied the intervention program. Clarify.
The participants participated in two orientation sessions about the experimental procedures, minimizing the learning effect during the study. However, the intervention program was previously tested as a pilot study? It is suggested to clarify.
The presentation of the results by the textual form and the tables presented is perceptible to the reader. The "Discussion" section integrates the results of the present research with previously published studies related to the object of study. In addition, explanations are provided for the reported results. It is suggested, however, that the result described below ("On the other hand, the present study did not detect any training-related differences between sexes in all assessed variables after a short-term PT performed on either firm or sand surfaces.") could be explained with another level of depth, from theoretical models, for example, since it is contrary to the hypothesis advanced in the introductory section. It could be an interesting discussion for readers.
Author Response
Referee 3
Main comments:
Congratulations to the authors for the topic under study. However, this review study presents some issues that may need to be addressed. In the comments below, some considerations are set out.
Author's response:
We thank the referee for his/her comments.
Comment 1
Although the title informs the variables under study it is necessary to consider the existence of a high variety of research methods. Once the title is of vital importance for the selection and reading of research reports, it is recommended to identify the report as to the research design used, being coherent with the methodological procedures described.
Author's response:
Thank you for your valuable suggestion. The title was changed as follows:
'' Effects of Surface-Type Plyometric Training on Physical Fitness in Schoolchildren of Both Sexes: A Randomized Controlled Intervention ''
Comment 2
The introductory section is well organized, including a contextualization of the problem under study, as well as a brief summary of the related state of the art. However, it is suggested that the rationale for the hypothesis ("We also hypothesized training-related differences between sex groups in the assessed variables after four weeks of PT.") could be structured in such a way as to inform the reader of its logic and consistency with a different depth, alluding to premises, concepts or theoretical models that are related to the object of study.
Author's response:
We thank very much the expert referee for his/her comment. The rationale for the hypothesis related to sex was developed as follows:
In addition to PT configuration, the subjects' characteristics, such as sex, should be considered to optimize training-related responses (Ramirez-Campillo et al., 2016). Previous studies suggested that the age from 8 to13 years can be considered as a sensitive period for the development of nervous and endocrine systems, as well as the corresponding anthropometric and physiological changes, which could affect the level of motor performance and their responses to learning and training stimuli (Almeida et al., 2021; Degache et al., 2010; Duzgun et al., 2011; Sunnegardh et al., 1988). Since sensitivity to PT programs may change differently according to sex over time/maturation (Peitz et al., 2018), we believe that it will be useful for practitioners to know whether sex could affect athletic performance following PT in prepubescent individuals. Specifically, few studies have examined sex-related effects following PT in pre - peak high velocity of children and did not detect any training-related differences between sexes in sprint, vertical and long jumps, and endurance (Marta et al. 2014; Skurvydas & Brazaitis, 2010; Vom Heed et al., 2007). Unfortunately, none of these studies compared the sexes' agility and dynamic balance responses. To the best of our current knowledge, no study has examined the effects of sex on physical fitness adaptations to surface-type PT in children.
Also the second hypothesis was changed as follows:
'' We also hypothesized no training-related differences between sex groups in the assessed variables after four weeks of PT. ''
Almeida, M.B.; Leandro, C.G.; Queiroz, D.D.R.; José-da-Silva, M.; Pessôa Dos Prazeres, T.M.; Pereira, G.M. das-Neves, G.S.; Carneiro, R.C.; Figueredo-Alves, A.D.; Nakamura, F.Y.; Henrique, R.D.S.; Moura-Dos-Santos, M.A. Plyometric training increases gross motor coordination and associated components of physical fitness in children. Eur. J. Sport Sci. 2021, 21, 1263-1272. doi: 10.1080/17461391.2020.1838620.
Degache F, Richard R, Edouard P, Oullion R, Calmels P. 2010. The relationship between muscle strength and physiological age: a crosssectional study in boys aged from 11 to 15. Ann Phys Rehabil Med 53: 180–188.
Duzgun I, Kanbur NO, Baltaci G, Aydin T. 2011. The effect of tanner stage on proprioception accuracy. J Foot Ankle Surg 50:11–15.
Sunnegardh J, Bratteby LE, Nordesjo LO, Nordgren B. 1988. Isometric and isokinetic muscle strength, anthropometry and physical activity in 8 and 13 years old Swedish children. Eur J Appl Physiol 58:291–297.
Peitz, M.; Behringer, M.; Granacher, U. A systematic review on the effects of resistance and plyometric training on physical fitness in youth- What do comparative studies tell us? PLoS One 2018, 13, 0205525. doi: 10.1371/journal.pone.0205525.
Comment 3
The "Participants" section is well described. However, since sample size plays an important role in the ability to make accurate inferences, was a statistical procedure performed to calculate the sample size? If so, it is suggested to report. Also, it seems unclear what sampling strategy was used. What was the procedure used for the participants to be allocated to the respective groups (experimental and control)? It is suggested to clarify.
Author's response:
We thank very much the expert referee for his/her comment.
- The sample size power has been already performed and included in the text. Please find the results of the statistical analysis section.
- '' Random allocation was maintained with a simple randomization stratified by sex, which resulted in the following assignments: two experimental groups [firm surface group (FG): performing PT on a firm surface, 27 girls and 25 boys; sand surface group (SG): performing PT on a dry surface of 20-cm deep sand, 24 girls and 21 boys)] and a control group (20 girls and 20 boys). ''
Please find changes in the text.
Comment 4
The success of an intervention program also depends on the coach/professor who applies it. In the manuscript it is not clear if it was the same coach/professor who applied the intervention program. Clarify.
Author's response:
We thank very much the expert referee for his/her comment. The following sentence was added at the end of the study design section:
'' The same research assistants conducted all testing procedures and the training program. ''
Please find changes in the text.
Comment 5
The participants participated in two orientation sessions about the experimental procedures, minimizing the learning effect during the study. However, the intervention program was previously tested as a pilot study? It is suggested to clarify.
Author's response:
We thank very much the expert referee for his/her comment.
- During the orientation session, the children were instructed on the proper technique for each training exercise, and their questions were answered to clear any doubts.
- The following changes are included in the training program section as follows:
'' No pilot study was conducted to test the intervention program. The length of the training period was selected in line with previous research that has reported the effectiveness of a 4-week PT program on SSC measures (Lloyd et al., 2012) and to allow for a test week before and after the training intervention within the educational term. Table 2 shows that the training program drills were based on findings from previous investigations (Lloyd et al., 2012) and incorporated the progressive overload principle by increasing the number of jumps and the levels of complexity according to previous PT guidelines (Lloyd et al., 2011).''
Comment 6
The presentation of the results by the textual form and the tables presented is perceptible to the reader. The "Discussion" section integrates the results of the present research with previously published studies related to the object of study. In addition, explanations are provided for the reported results. It is suggested, however, that the result described below ("On the other hand, the present study did not detect any training-related differences between sexes in all assessed variables after a short-term PT performed on either firm or sand surfaces.") could be explained with another level of depth, from theoretical models, for example, since it is contrary to the hypothesis advanced in the introductory section. It could be an interesting discussion for readers.
Author's response:
We thank very much the expert referee for his/her comment. The discussion related to sex findings was rewritten as follow:
On the other hand, the present study did not detect any training-related differences between sexes in all assessed variables after a short-term PT performed on either firm or sand surfaces. These findings corroborate the results of previous reports conducted with prepubescent children, which reported no significant differences in PT response related to sex [1,17,18,58], suggesting that the trainability of plyometrics in children appears not to be affected by sex-specific effects [4]. Regardless of maturity levels, children in this study attend the same school and participate in the same physical education programs. It has been reported that the physical fitness responses to training could be potentially related to the biological maturation in body size, shape, and composition [1]. Despite the maturation process is earlier in boys, girls gain around 3.5 kg of muscle mass annually and accumulate a higher amount of fat mass during PHV, while boys gain much more muscles (7.2 kg per year) (Tonnessen et al. 2015). These concurrent processes can disproportionately reduce the strength in girls (Romero et al., 2021). Unfortunately, the current study lacks more physiological and biomechanical assessments to better understand the underlying mechanisms of plyometric training-induced adaptations in both sexes. Information relating to these factors may help practitioners to optimize well-rounded scholar-training programs at a time of rapid changes such as the prepubertal growth spurt (Marta et al., 2014).
Tønnessen E, Svendsen IS, Olsen IC, Guttormsen A, Haugen T. Performance development in adolescent track and field athletes according to age, sex and sport discipline. PLoS One 10: e0129014, 2015.
Romero C, Ramirez-Campillo R, Alvarez C, Moran J, Slimani M, Gonzalez J, Banzer WE. Effects of Maturation on Physical Fitness Adaptations to Plyometric Jump Training in Youth Females. J Strength Cond Res. 2021 Oct 1;35(10):2870-2877.
Round 2
Reviewer 2 Report
The authors have followed the reviewers suggestions and improved the manuscript. There are however some amendments which warrant attention.
Simple Summary: As for the previous suggestions, it would be appropriate to modify the recommendation for sand training.
Line 77, “differently” should be removed or moved before “affect” in line 76.
Line 120 “with 48 of rest”, 48 of what?
Line 121 “With the both” please remove “the”
Line 121 procedures
Line 140 modify was with were.
Line 369, if the authors state “might be “”more”” related to”, the sentence should end with another sentence starting with “than”. In this case a rewording is necessary.
Thanks
Author Response
============================================================
REVIEW BIOLOGY
Minor Revisions
============================================================
Manuscript ID number: biology-1793016
Dear Editor and Referees:
Enclosed is the revision of the manuscript ID number: biology-1793016 entitled "Effects of Surface-Type Plyometric Training on Physical Fitness in Schoolchildren of Both Sexes ". Thank you very much for the opportunity to revise our manuscript, as well as for the valuable and helpful comments and suggestions. We do believe that the paper has significantly improved after this revision. We have modified the manuscript according to all comments and suggestions raised by the referees.
Round 2
Referee 2
Main comments:
The authors have followed the reviewers suggestions and improved the manuscript. There are however some amendments which warrant attention.
Author's response:
We thank the expert reviewer for his/her comments. The comments were addressed in specific points.
Comment 1
Simple Summary: As for the previous suggestions, it would be appropriate to modify the recommendation for sand training.
Author's response:
Thank you for your valuable comment.
Recommendations for sand surfaces were modified as follows :
'' Since many schools do not have access to sand surfaces, firm surface remains suggestible when PT is envisaged for schoolchildren. ''
Comment 2
Line 77, “differently” should be removed or moved before “affect” in line 76.
Author's response:
Thank you for your valuable comment.
''Differently'' was moved before ''affect''. Please find changes in the text.
Comment 3
Line 120 “with 48 of rest”, 48 of what?
Author's response:
We thank the expert referee for his/her comment.
with 48 hours of rest. Please find changes in the text.
Comment 4
Line 121 “With the both” please remove “the”
Author's response:
We thank the expert referee for his/her comment.
''the'' was removed. Please find changes in the text.
Comment 5
Line 121 procedures
Author's response:
We thank the expert referee for his/her comment.
''s'' was added. Please find changes in the text.
Comment 6
Line 140 modify was with were.
Author's response:
We thank the expert referee for his/her comment.
''was'' was modified by ''were''. Please find changes in the text.
Comment 7
Line 369, if the authors state “might be “”more”” related to”, the sentence should end with another sentence starting with “than”. In this case a rewording is necessary.
Author's response:
We thank the expert referee for his/her comment.
''more'' was removed. Please find changes in the text.